# Influences of COVID-19 Work-Related Fears and Anhedonia on Resilience of Workers in the Health Sector during the COVID-19 Pandemic

Alexander Maget, Melanie Lenger *, Susanne A. Bengesser, Armin Birner, Frederike T. Fellendorf, Eva Fleischmann, Jorgos N. Lang, Martina Platzer, Robert Queissner, Michaela Ratzenhofer, Elena Schönthaler, Adelina Tmava-Berisha, Robert M. Trojak, Nina Dalkner and Eva Z. Reininghaus

Clinical Department of Psychiatry and Psychotherapeutic Medicine, Medical University Graz, Auenbruggerplatz 31, 8036 Graz, Austria
* Correspondence: melanie.lenger@medunigraz.at; Tel.: +43-316-385-81755

**Abstract:** Background: During the peak of the COVID-19 pandemic, healthcare workers worked under stressful conditions, challenging their individual resilience. Therefore, we explored the bidirectional influence of resilience and the factors of COVID-19 work-related fears and anhedonia in Austrian healthcare workers. Methods: Healthcare workers in Austria completed an online survey at two points in time. The first measurement started in winter 2020/2021 ($t$1), and a second measurement began approximately 1.5 years later ($t$2). One hundred and eight six individuals completed both surveys and were investigated in a longitudinal design. We applied the Resilience Scale, the Snaith-Hamilton Pleasure Scale, and a self-created questionnaire assessing COVID-19 work-related fears. We used a repeated measures analysis of variance and applied Pearson-Correlations as well as univariate and multivariate analyses of covariance. Results: Resilience was significantly correlated with COVID-19 work-related fears and anhedonia at both points in time in all participants. We found no significant differences for frontline vs. non-frontline workers at $t$1 and $t$2. Resilience decreased significantly over time. Limitations: Most subjects were examined cross-sectionally. Frontline workers were underrepresented in our sample. Conclusion: Our findings highlight the importance of resilience in healthcare providers. Steps must be taken to maintain and promote resilience in healthcare workers. We suggest that the improvement of resilience, dealing with fears and uncertainty, and the ability to experience joy might have a beneficial influence on the respective other categories as well.

**Keywords:** resilience; healthcare worker; COVID-19 pandemic; mental health; anhedonia

## 1. Introduction

The upcoming of the coronavirus disease (COVID-19) pandemic hit the world hard, challenging political, economic, and health care systems. COVID-19 is an airborne, infectious disease caused by the severe acute respiratory syndrome coronavirus 2 (SARS-CoV-2). Affected patients suffer from fever, cough, and sore throat, but also atypical pneumonia and severe acute respiratory syndrome, which requires intensive medical care including artificial ventilation, Struyf et al. (2020). Due to its high virulence, it spread rapidly all over the world, being labeled a global pandemic by the World Health Organization on 12 March 2020. With causal treatment and vaccination unavailable, governments had to resort to measures like lockdowns, travel restrictions, and the implementation of social distancing rules to counter the unchecked spread of the virus. In Austria, the first lockdown was imposed on 10 March 2020 to prevent hospitals and other health care facilities from reaching their maximal capacities. Further lockdowns in the autumn of 2020 and during the year 2021 greatly impacted the daily life of workers in the healthcare sector (WHS).

While whole industries were threatened by bankruptcy caused by lockdowns, leaving thousands unemployed or finding themselves in forced inactivity; health care workers had

to carry a substantial part of the burden brought onto society. Beside stressful working conditions, health care professionals had to deal with the imminent threat of infecting themselves and people nearby with the disease. Therefore, we aimed to determine the factors influencing individual resilience under such stressful conditions. In particular, we investigated the bidirectional connection between resilience, COVID-19 work-related fears, and anhedonia within WHS. While previous studies focused on the impact of resilience within psychiatric patients (Dalkner et al. 2021; Fleischmann et al. 2022; Schönthaler et al. 2022; Min et al. 2013), patients with chronic diseases (Buscetta et al. 2022; Kim et al. 2019; Sbragia et al. 2022; Tso et al. 2020), and cancer Seiler and Jenewein (2019), as well as child development, Masten and Reed (2002), the application of resilience concepts to the workplace is relatively new. Without the burden of a pandemic situation, King et al. (2016) had already discussed the necessity of including resilience, a thus far overlooked concept into today's organizations.

Resilience, as a theoretical psychological concept has been changing in its definition throughout its investigation. Although nowadays there are three different approaches, defining resilience as either a personality trait, a process, or an outcome, most definitions are based around the two concepts of adversity and positive adaption, Fletcher and Sarkar (2013). According to the authors, its underlying factors can be divided into protective factors, such as hardiness, positive emotions, extraversion, self-efficacy, spirituality, positive affect, and promotive factors, which implies different processes of maintenance and interventive measures of resilient cognition and behavior. However, Leppert et al. (2008) also take note of a binary approach in which their discussed resilience scale (RS-25 and its short version RS-13) can be divided in the sub-scales "Acceptance of Self and Life" and "Personal Competence".

The process perspective that draws from developmental psychology understands resilience as a process of successful adaption during strain, being thus theorized as a more dynamic construct that is alterable during lifetime, according to Gawlytta and Rosendahl (2015). This is the reason why research in process approach is usually focused on the identification of criteria for good development conditions and properties of persons which have an influence on the resistance abilities of the individual. On the other hand, the differential psychological trait perspective considers resilience as a stable personality trait in the sense of a psychological resistance ability often called "trait-resilience". The current pandemic situation may have caused an increased rate of fears (e.g., fear of infection, fear of job loss, etc.) that could be more easily coped with by people with a higher trait-resilience score, as they tend to stand out in their flexible adaptivity, their ability to reflect, and their social competence, whereas less resilient persons are more likely characterized as less flexible and more disconcerted by changes, making them more prone to fear and stress.

They also take note of the fact that resilience has often been theorized as a multidimensional construct that includes constitutional personality traits as well as competences of coping with strain, Gawlytta and Rosendahl (2015). Nevertheless, there is a noticeable decrease of resilience during the COVID-19 pandemic, as a longitudinal study of Kimhi et al. (2020) suggests.

In addition, resilience is being discussed as a modulating protective process in somatic diseases and as a regulator in pain and stress perception, Leppert et al. (2008).

As there is strong evidence to both genetic and cultural differences, Ungar (2008) in resilience, more recent theoretical approaches discuss resilience as a multimodal dynamic process, such as the Multi-System of Resilience by Liu et al. (2017). As an integrative approach, the discussed model includes interpersonal resilience factors, socio-ecological factors that are organized concentrically around core resilience. In this model, a stressful situation like a pandemic would be considered an external factor that influences the different systems (e.g., in the family with different stressors or in the personal biological factors e.g., by suffering from a post-COVID-19 syndrome) Liu et al. (2017).

Especially during the actual pandemic situation that led to social distancing, actual protective elements of mental health promotion, like qualitative social contacts or spare

time, may have been more difficult to maintain than before the pandemic. Thus, resilience may be an important factor to take note of and help to surpass a threatening strain. Apart from these external factors of resilience like social environment and support, job satisfaction, and supportive working conditions, McCann et al. (2013), there are internal factors such as self-efficacy, positive emotions, and coping strategies (McCann et al. 2013; Çam and Büyükbayram 2017).

As such, the ability to experience joy and pleasure is an important feature and resource within resilience, rendering anhedonia of special interest when investigating influencing factors of resilience.

Anhedonia, as the clinical psychological term for a lack of pleasure or interest, is strongly associated with chronic stress and depression, especially in susceptible but not resilient individuals with a strong neurobiological component, Prakash et al. (2020). According to the authors, the symptom that reflects an impaired brain reward function and is also modulated by serotonergic plasticity in the dorsal raphe nucleus in rats can be considered a debilitating symptom of several psychiatric disorders, including major depressive disorder.

Further neurobiological substrates of anhedonia and resilience were shown by D'Addario et al. (2021). During the COVID-19 pandemic, there was a noticeable increased tendency of anhedonic behavior as well as dysfunctional coping. For instance, Landaeta-Díaz et al. (2021) take note of an increase of palatable foods as a strategy to mitigate negative emotions, such as anxiety. Also, individuals that had been infected by COVID-19 present higher fatigue ad anhedonia rates, El Sayed et al. (2021). Furthermore, the pandemic situation also had an impact on non-COVID-19 work-related medical treatments. For example, mothers in postpartum period presented higher anhedonia rates during the pandemic than before, emphasizing the importance of healthcare during the pandemic, Zanardo et al. (2020).

Further, Schönthaler et al. (2022) showed COVID-19 fears to be elevated in clinical and non-clinical samples in times of higher infection and mortality during the pandemic.

We therefore hypothesized that there are individuals that present more psychological vulnerabilities during the pandemic than others that may cause greater emotional distress, somatization, and reduction of sleep quality, also modulated by COVID-19 work-related fears. Resilience can serve as an important protective process during such times of social distancing and fears about health or economic problems of individuals and their peers (e.g., family and friends). Therefore, more individual vulnerability might suggest less resilience during a pandemic.

As the actual crisis is a situation that intensively affects WHS, we aimed to explore the psychological well-being of this specially strained group in order to detect certain vulnerability factors and changes over time. The pandemic situation has challenged health care workers on a daily basis, with an increased amount of distress at the workplace that came along with personal worries from the immediate microsystem of the workers (e.g., job worries or COVID-19-infection of family and spouses). Current works suggest that WHS during the pandemic present a heightened risk of mental health problems, like psychological distress, insomnia, alcohol or drug abuse, symptoms of post traumatic stress disorder, depression, anxiety etc. that are predicted by organizational, social, personal, and psychological factors, Stuijfzand et al. (2020). Other papers corroborate an increase of depressive symptoms, anxiety, and lower levels of proactive coping, Pearman et al. (2020). According to Braquehais et al. (2020), the higher prevalence of anxiety and depressive symptoms is modulated by COVID-19 exposure, epidemiological issues, material resources, human resources, and personal factors.

Subsequently, we hypothesized that resilience would decrease over time, with frontline WHS being more severely affected than non-frontline WHS. We further expected anhedonia and COVID-19 work-related fears to be associated with resilience in our sample.

## 2. Materials and Methods

### 2.1. Study Design

The study was conducted at the Clinical Department of Psychiatry and Psychotherapeutic Medicine at the Medical University of Graz in Austria. It was conducted according to the Declaration of Helsinki and was approved by the Ethics Committee of the Medical University Graz (EK-number: 32 329 ex 19/20). The ethics committee agreed to the project with the project title "Psychosocial interests on the SARS-CoV-2 pandemic on employees of healthcare in Austria".

During the COVID-19 pandemic 2020, healthcare workers in Austria were asked to complete an online survey via LimeSurvey (https://www.limesurvey.org/de/, accessed on 21 January 2022) at two points in time. Data collection for the first measurement started in winter 2020/2021 (*t*1) during the second lockdown in Austria. The total lockdown in Austria by a declaration of the Austrian Federal Ministry of Social Affairs, Health, Care and Consumer Protection and included the total closure of shops, schools, and universities. Other measures taken were social distancing and recommended home-office. Wearing a face mask was mandatory at all public buildings. During data collection for *t*1, the maximal total incidence in Austria was 545 with 76.554 active cases and 567 inpatients in intensive care (Nationale Referenzzentrale für Influenza-Epidemiologie 2020).

A second measurement was done approximately 1.5 years later, beginning in winter 2021/2022. At this time, the B.1.1.529 variant, mostly known as the "Omicron" variant had already been spreading in Austria. In November 2021, a lockdown had been declared that lasted until the 14th of December 2021. Maximal infection states during that time were an incidence of 1005, 157.349 active cases, and a maximum of 623 inpatients in intensive care (Nationale Referenzzentrale für Influenza-Epidemiologie 2020) Again, the link for the online survey was sent out via work council departments and clinic managements. Anonymous data collection during both examinations was ensured with anonymized participant codes.

### 2.2. Materials

To measure resilience, we applied the short version of the Resilience Scale designed by Wagnild and Young (1993) in the German version by Leppert et al. (2008). The short version RS-13 consists of 13 7-point Likert Scale items (1 being no agreement and 7 being total agreement) that results in a total score [Total Resilience] with a maximum of 91 points. According to the authors, it can be further divided in the two sub scales "Acceptance of Self and Life" [Acceptance] and "Personal Competence" [Competence]. Higher scores represent higher resilience. The questionnaire presents high internal consistency with $\alpha = 0.90$.

For the anhedonia measurement, we applied the German translation of the Snaith-Hamilton Pleasure Scale (SHAPS) by Snaith et al. (1995) provided by Franz et al. (1998). The questionnaire depicts anhedonia with 15 Likert-4 scaled Items (0 being no and 3 being total agreement), with lower scores displaying more symptoms of anhedonia.

For fears related to the pandemic we created a questionnaire in German, assessing COVID-19 work-related fears in 6 different parts. The 1-item question gave the possibility to mark up to five different fears. In addition, due to the new situation that may create many different possible worries, open answers were also possible. The participants could mark whether they presented the related fears or not. The COVID-19 work-related fear score of each participant consisted of the total sum of fears they presented (with minimum 0 and maximum 5 COVID-related fears). The applied items of the questionnaire can be found in Table 1.

**Table 1.** Self-constructed COVID-19 work-related fears questionnaire.

| Are the actual developments around the SARS-CoV-2 pandemic scaring you in any way? | Yes, I am fearing a potential economic crisis. Yes, I am worrying about my health. Yes, I am worried about the health of my relatives. Yes, I am fearing for my job. Yes, for other reasons. No |
|---|---|

### 2.3. Participants

At first, we aimed to explore different prevalence rates at two different times of the pandemic, which we measured cross-sectionally. Concordance of participants at both examinations was low (N = 186), so the two samples were interpreted individually.

For both points in time, we aimed to unveil differences in resilience, anhedonia, and COVID-19 work-related fears according to individual differences in the health care system, as well as age, sex, type of work (e.g., physician or nurse), and the difference between frontline WHS, who were having direct contact with COVID-19 infected patients, and non-frontline WHS.

### 2.4. Statistical Analyses

We analyzed the whole dataset for the main parameters "Resilience", "Anhedonia" and "COVID-19 work-related fears". After excluding those participants who did not give information about their age or sex, we were able to recollect a sample of 1367 subjects for $t1$ and 1181 subjects for $t2$.

Furthermore, we applied statistical inferences with subjects that participated both times (N = 186). We used a repeated measures analysis of variance to test for changes over time in resilience. Thus, in addition to the descriptive analyses, we applied Pearson-Correlations as well as univariate and multivariate Covariance Analyses (ANCOVA and MANCOVA) to search for group differences at both points in time, respectively. We controlled for age and gender in all analyses.

### 3. Results

*3.1. Descriptive Statistics of Resilience, Anhedonia and COVID-19 Work-Related Fears in WHS at t1 and t2*

The mean score of the resilience scale (RS13) was 74.29 (SD = 11.52) at $t1$, which can be categorized as "high" and 71.58 (SD = 12.76) at $t2$, which can be categorized to be "moderate", according to Leppert et al. (2008). The means of the SHAP Scale ($t1$: mean = 35.96 SD = 7.48; $t2$: mean = 37.45 SD = 6.57) and the means of COVID-19 work-related fears ($t1$: mean = 1.66 SD = 0.83; $t2$: mean = 0.92 SD = 1.05) where used for further analyses.

*3.2. Correlation between Resilience Anhedonia and COVID-19 Work-Related Fears at t1 and t2*

The partial correlation (controlled for age and sex) showed that resilience was significantly positively correlated with COVID-19 work-related fears at $t1$ and negatively at $t2$ in all WHS (see Table 2).

Furthermore, resilience was significantly correlated with anhedonia at both points in time in WHS (partial correlation; controlled for age and sex; see Table 3). However, these correlations did not differ between frontline WHS and non-frontline WHS.

**Table 2.** Correlations of COVID-related fears with the resilience scales at.

| | | $t1^{1}$ ($df = 103$) | | |
| --- | --- | --- | --- | --- |
| | | Total Resilience [2] | Competence | Acceptance |
| | $r$ | 0.271 | 0.263 | 0.222 |
| | sig. | 0.005 ** | 0.007 ** | 0.023 ** |
| | | $t2^{1}$ ($df = 1169$) | | |
| COVID-related fears | | Total Resilience [2] | Competence | Acceptance |
| | $r$ | −0.137 | −0.120 | −0.145 |
| | sig. | 0.000 ** | 0.000 ** | 0.000 ** |

Note. [1] After controlling for sex and age. [2] RS13-Scores = total score of the resilience scale. ** $p < 0.01$.

**Table 3.** Correlations of SHAPS with the resilience scales.

| | | $t1^{1}$ ($df = 103$) | | |
| --- | --- | --- | --- | --- |
| | | Total Resilience [2] | Competence | Acceptance |
| | $r$ | 0.536 | 0.517 | 0.443 |
| | sig. | 0.000 ** | 0.000 ** | 0.000 ** |
| | | $t2^{1}$ ($df = 1169$) | | |
| SHAPS | | Total Resilience [2] | Competence | Acceptance |
| | $r$ | 0.374 | 0.371 | 0.315 |
| | sig. | 0.000 ** | 0.000 ** | 0.000 ** |

Note. [1] After controlling for gender and age. [2] The total resilience scale consists of the competence and acceptance items. SHAPS = Snaith-Hamilton Pleasure Scale. ** $p < 0.01$.

*3.3. Frontline WHS vs. Non-Frontline WHS at t1 and t2*

Two MANCOVAs (controlled for age and sex) showed no significant differences for frontline vs. non-frontline WHS at *t*1 (RS13-Total: (F(5.872) = 0.044, $p = 0.834$, η2 = 0.000), RS13-Competence: (F(11.288) = 0.177, $p = 674$, η2 = 0.000), RS13-Acceptance: (F(.877) = 0.047, $p = 0.829$, η2 = 0.000)) and *t*2 (RS13-Total: (F(62.36) = 0.383, $p = 0.536$, η2 = 0.000), RS13-Competences: (F(62.36) = 0.758, $p = 0.384$, η2 = 0.000), RS13-Acceptance: (F(1) = 0.00, $p = 0.997$, η2 = 0.000)) for total resilience as well as resilience sub scales.

*3.4. Repeated Measure Analyses for a Subsample at t1 Compared to t2 (n = 186)*

An ANCOVA (controlled for age and sex) showed no significant main effect for frontline WHS and non-frontline WHS (F(91.56) = 0.453, $p = 0.592$, η2 = 0.002) but a significant main effect for winter 2020/2021 vs. winter 2021/2022 (F(313.56) = 6.508, $p = 0.012$, η2 = 0.034). The interaction was not significant (F(3.227) = 0.067, $p = 0.796$, η2 = 0.000). Please see Figure 1 for the results of the ANCOVA.

For the subsample, all correlations between resilience and anhedonia are comparable to the correlations in the whole sample. For more details, please see Table 4.

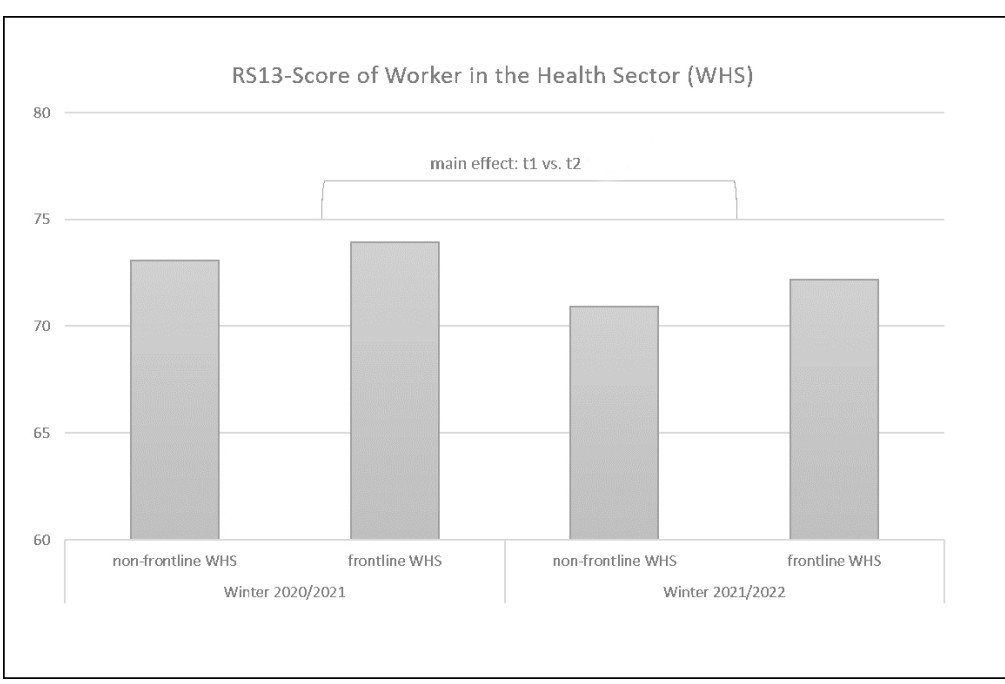

**Figure 1.** Repeated Measures ANCOVA (controlled for age and sex) in RS13-Scores for frontline vs. non-frontline WHS in 2020/2021 and 2021/2022.

**Table 4.** Correlations between SHAPS and total resilience at both measurement points.

|  |  | SHAPS *t1* | SHAPS *t2* | RS13-Score *t1* | RS13-Score *t2* |
|---|---|---|---|---|---|
| SHAPS *t1* | *r* | 1 | 0.529 [1] | 0.365 [1] | 0.187 [2] |
|  | sig. | - | 0.000 | 0.000 | 0.011 |
|  | *N* | 1362 | 187 | 1362 | 186 |
| SHAPS *t2* | *r* |  | 1 | 0.216 [1] | 0.363 [1] |
|  | sig. | - | - | 0.003 | 0.000 |
|  | *N* |  | 1138 | 187 | 1137 |
| Total Resilience *t1* | *r* |  |  | 1 | 0.616 [1] |
|  | sig. | - | - | - | 0.000 |
|  | *N* |  |  | 1362 | 186 |
| Total Resilience *t2* | *r* |  |  |  | 1 |
|  | sig. | - | - | - | - |
|  | *N* |  |  |  | 1137 |

Note. [1] The correlation is significant on a two-tailed 0.01 level. [2] The correlation is significant on a two-tailed 0.05 level.

## 4. Discussion

Our finding that resilience was positively correlated to COVID-19 work-related fears at the beginning of the pandemic is intriguing. Given the uncertainty of the situation and its development during a virus pandemic, a certain amount of caution and worries might have posed as a reasonable strategy in dealing with a possible threat. As knowledge and measures to deal with this threat broadened, fears became more and more debilitating instead of reasonable. At *t2*, resilience proved to be negatively associated with COVID-19 work-related fears, which is in accordance with previous publications. Setiawati et al. (2021) could show anxiety and resilience scores to be negatively correlated in Indonesian health care workers during the COVID-19 pandemic. In line with this, Rayani et al. (2022) found a weak but significant association between resilience and COVID-19 anxiety in Iranian healthcare workers during the pandemic. We could, therefore, enrich the scientific data suggesting a strong connection between those factors in WHS for the COVID-19 pandemic. However, there are also hints that this connection still holds true without the situation

of a global pandemic. Before COVID-19, Hjemdal et al. (2011) could show a connection between resilience and anxiety in Norwegian adolescents. They found not only anxiety but also depression scores to be higher within subjects with low resilience scores. Furthermore, Shi et al. (2015) found a negative correlation between resilience and anxiety in Chinese medical students.

In contrast to previous studies (Awano et al. 2020; Baskin and Bartlett 2021), we found no significant difference in resilience between frontline and non-frontline WHS. Combined with our finding of decreasing resilience in all WHS in our sample, this might suggest stressors in the healthcare system beyond fears and health concerns which were aggravated by the pandemic affecting all personnel responsible for the continuation of health care under such strenuous conditions.

In our study, we could show resilience scores to be associated with the ability to experience pleasure and joy. We therefore suspect a bidirectional connection between factors necessary to cope with and recover from hardship and influences changing a person's perception of joy.

Literature regarding the association between resilience and anhedonia is scarce. While there are some studies which investigated resilience and anhedonia in animal models (Prakash et al. 2020; Gaspar et al. 2021), research in humans is, to our best knowledge, currently missing. This might be due to the fact that anhedonia poses as a key criterion for depression and as such is subsumed in studies investigating the relationship between resilience and depression. Still, it might be an interesting factor, as the ability to experience joy during depressive episodes could pose as a source of resilience in terms of "bouncing back" from depressive perception and experience. In line with that, patients with atypical depression, a form of depression with which patients are still responsive to positive events, could be shown to have shorter episodes during the course of their illness, Nierenberg et al. (1998). Li et al. (2020) showed resilience to have positive effects not only on mood, but also on deterioration of mood due to negative life events in 608 Chinese college students. Further, resilience was negatively related to depressive mood in a study investigating 1076 Korean employees, Shin et al. (2019). Focusing more on the aspect of resilience as being a resource to deal with mood deterioration in an effective way, than it representing a preemptive protection, Smith (2009) showed high resilience in depressed elderly African Americans to increase their probability to seek professional help. Another common aspect of depression is hopelessness, Schneider et al. (2017). Hjemdal et al. (2012) found resilience to predict lower levels of hopelessness in 532 healthy subjects, independent from mood, stressful life events, and personality.

While human studies are currently missing, Heshmati and Russo (2015) tried to identify the neurological circuits behind resilience to anhedonia in rodents. They found one third of mice to be resilient to the development of anhedonia due to the engagement of active counter-regulatory mechanisms like beta-catenin-mediated microRNA regulation, Dias et al. (2014) and DNA methylation of the Crf gene, Elliott et al. (2010). Further, Christensen et al. (2011) found stress resilient rats to be clustered with rats recovering from anhedonia due to antidepressant treatment, suggesting parallels between antidepressant effectiveness and functions of resilience. They also identified dj842G6.1.1, Nr4a3, and Sstr2 as genes associated with stress-resilience.

Some studies tried to elucidate the neuro-endocrinological mechanisms underlying the relationship between resilience and depression in humans. The hypothalamic-pituitary-adrenal (HPA) axis is the central modulating unit for stress regulation in our body and, as such, the key component for stress resilience. Through its effects on limbic, mid brain, and prefrontal structures, it influences our behavioral responses to stress, Russo et al. (2012). Neuropeptide Y and dehydroepiandrosterone could be identified as agents regulating HPA activity (Russo et al. 2012; Morgan et al. 2002; Rasmusson et al. 2003) by exerting neuroprotective effects.

Resilience was found to be generally lower in people suffering from psychiatric disorders, Shrivastava and Desousa (2016). This is of interest, as distress could trigger

affective episodes in patients with bipolar disorder, Fellendorf et al. (2021). Nevertheless, as exposure to adverse and stressful experiences is necessary for resilience to develop, Rutter (1987), considerations about the reciprocal connection between resilience and psychosocial well-being arise. The development of resilience as a factor protecting individuals from the psychological impact of stressful life events could be seen as a kind of training effect. An individual has to experience adversity to learn to cope with it in a proper manner, but has to be protected from overwhelming psychological trauma, which would cause long-lasting damage and impede the evolution of healthy coping strategies, especially during the developmental phases during childhood and adolescence. Similar to this, Meichenbaum postulated the stress inoculation concept, Meichenbaum (2017), which aims to improve stress resistance in individuals through individual therapy to develop and apply skills to deal with acute and chronic stress. It consists of education, skill acquisition, and application. As experience is an important factor within internal resilience, McCann et al. (2013), this approach can be understood as a controlled setting to build resilience.

Our findings that resilience scores decreasing over the course of the pandemic are in accordance with this, and alarming. While resilience is an important factor in dealing with distress, the intensity and duration of straining and stressful conditions can seemingly reach a dimension where an individual's capability to benefit from their resilience is exhausted.

This is especially important, as we found resilience scores to be initially generally high throughout all subgroups in our sample. This may be contributed to health-care professions being psychosocially demanding and stressful, Santos et al. (2010), creating a selection process after which only people with generally high resilience choose to stay in this field of work. Still, even these individuals were considerably affected by more than a year of working under high-stress conditions, after which their scores had dropped to a mediocre rating. These findings highlight the importance of resilience in health care providers, especially during a pandemic. Therefore, steps need to be taken to maintain and promote personal resilience in health care workers. Our findings suggest that the improvement of resilience, dealing with fears and uncertainty, and the ability to experience joy and pleasure might have a beneficial influence on the other categories as well. We therefore suggest implementing strategies to further such skills in people working in the health care system to profit of the probable synergy. Such interventions should include enhancing people's self-efficacy and mindfulness, Rees et al. (2015), conveying methods for relaxation and delay of judgement of stressful situations, Sood et al. (2011) using mostly cognitive-behavioral approaches, Robertson et al. (2015).

*Limitations*

There were several limitations to our study. Most subjects were examined cross-sectionally, which allows no interpretation of the parameters during the course of time, as study samples were not comparable. Still, we were able to investigate 186 subjects over the course of 1.5 years, allowing us to draw conclusions about the development of resilience, anhedonia, and COVID-19 work-related fears over time. Lastly, frontline WHS made up 22.6% of all participants, meaning they were underrepresented in our sample. However, with the health care system including a vast variety of fields and professions, it is to be expected that only a fraction of health care providers is in contact with COVID-19 patients.

**5. Conclusions**

In this study, we hypothesized that resilience would decrease over time, with frontline WHS being more severely affected than non-frontline WHS. We further expected anhedonia and COVID-19 work-related fears to be associated with resilience in our sample. While we found no difference between frontline and non-frontline WHS, resilience scores decreased over time, and anhedonia as well as COVID-19 work-related fears were correlated with resilience.

Our findings show in an alarming way that stress and overburdening due to a plurality of pandemic-associated factors impair the individual and systemic capacity of people

working in the healthcare sector and the healthcare system as a whole. Furthermore, they highlight the importance of resilience in healthcare providers. Steps must be taken to maintain and promote resilience in WHS. We suggest that the improvement of resilience, dealing with fears and uncertainty, and the ability to experience joy might have a beneficial influence on the respective other categories as well.

**Author Contributions:** Conceptualization, A.M. and M.L.; methodology, A.M., M.L., J.N.L. and N.D.; validation, A.M., M.L., S.A.B., A.B., N.D., F.T.F., E.F., J.N.L., M.P., R.Q., M.R., E.S., A.T.-B., R.M.T. and E.Z.R.; formal analysis, M.L. and A.M.; investigation, A.M., M.L., N.D., F.T.F., E.S. and E.Z.R.; resources, A.M., M.L., N.D., F.T.F., E.S. and E.Z.R.; data curation, A.M., M.L., N.D., F.T.F., E.S. and E.Z.R.; writing—original draft preparation, A.M. and M.L.; writing—review and editing, A.M., M.L., S.A.B., A.B., N.D., F.T.F., E.F., J.N.L., M.P., R.Q., M.R., E.S., A.T.-B., R.M.T. and E.Z.R.; visualization, M.L. and J.N.L.; supervision, N.D. and E.Z.R.; project administration, M.L.; funding acquisition, S.A.B. All authors have read and agreed to the published version of the manuscript.

**Funding:** The COVID-19 Research at the Clinical Department of Psychiatry and Psychotherapeutic Medicine was funded by the Austrian Science Fund (FWF KLI 968) as well as Land Steiermark.

**Institutional Review Board Statement:** This study was conducted according to the Declaration of Helsinki and was approved by the Ethics Committee of the Medical University Graz (EK-number: 32 329 ex 19/20). The ethics committee agreed to the project with the project title "Psychosocial interests on the SARS-CoV-2 pandemic on employees of healthcare in Austria".

**Informed Consent Statement:** Informed consent was obtained from all subjects involved in the study.

**Data Availability Statement:** Data are stored at the Clinical Department of Psychiatry and Psychotherapeutic Medicine of the Medical University Graz and can be provided upon request by the corresponding author.

**Acknowledgments:** We would like to thank all participants of our study. Open Access Funding by the Austrian Science Fund (FWF).

**Conflicts of Interest:** The authors declare no conflict of interest. The funders had no role in the design of the study; in the collection, analyses, or interpretation of data; in the writing of the manuscript; or in the decision to publish the results.

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
