# Peer review of "Influences of COVID-19 Work-Related Fears and Anhedonia on Resilience of Workers in the Health Sector during the COVID-19 Pandemic"

_socsci, doi:10.3390/socsci11120578_

Round 1

Reviewer 1 Report

Comment 1:

It would be preferable to combine lines 76 to 81 into one paragraph. In addition, you can elaborate on the multidimensional concepts of resilience. Explain how both external and internal protective factors influence resilience. Ensure that each paragraph is connected to the following one.

Comment 2:

It would be preferable to include lines 112-113 in the following paragraph.

Comment 3:

Instead of comparing the results of studies on adolescents, it is preferable to search for suitable comparisons for healthcare workers. (line 293-295)

Comment 4:

In the conclusion, you should restate the findings in accordance with the research questions.

Author Response

Honored colleague,

Thank you for your important constructive feedback.

We have adjusted our manuscript according to your suggestions and hope you are satisfied with the updated version. All changes in the manuscript are held in green to make them easier to find.

Comment 1 - Combined and shortened the sentences and added a paragraph on internal and external resilience factors while better connecting the paragraphs regarding content.

Comment 2 - Switched the sentence to the following paragraph.

Comment 3 - Added a reference with adult study subjects. As pre-pandemic literature on anxiety and resilience is relatively scarce we abstained from removing the existing reference and hope this is agreeable to you.

Comment 4 - Added a paragraph subsuming our hypotheses and results.

We would like to thank you again and hope to have met your expectations with our corrections.

Reviewer 2 Report

This study aimed to explore the bidirectional influence of resilience and the factors COVID-19 work-related fears and anhedonia in Austrian healthcare workers.

The design of the study and methodology were chosen adequately to address the objectives. Healthcare workers in Austria completed an online survey at 9 two points in time. First measurement started in winter 2020/2021 (t1), a second measurement ap-10 proximately 1.5 years later (t2). 186 individuals completed both surveys and were investigated in a 11 longitudinal design.

The results clearly presented. Resilience was significantly correlated with COVID-19 work-related fears and 15 anhedonia at both points in time in all participants. The authors found no significant differences for frontline 16 vs. non-frontline workers at t1 and t2. Resilience decreased significantly over time.

Conclusion. The findings highlight the importance of resilience in healthcare providers. Steps must be taken to maintain and promote resilience in healthcare workers. The authors suggest that the improvement of resilience, dealing with fears and uncertainty, and the ability to experience joy might have a beneficial influence on the respective other categories as well.

A strong part of the study was good methodology and the design of the study. The weaker part of the study was that the most subjects were examined cross-sectionally. Frontline workers were underrepresented in their sample.

The manuscript is suitable for printing. It is recommended to print without corrections.

Author Response

Honored colleague,

We thank you for your kind feedback and are happy that you are satisfied with the quality and content of our paper.

Reviewer 3 Report

The article deals with an interesting issue related to resilience at work. The problem is well supported by data from the literature. The study design was properly planned and conducted. I have not found information about the approval of research by the ethics committee. The presentation of the results in the tables is correct. Figure 1 is not clear when it comes to marking the main effect. The interpretation of the results indicates deficiencies. I do not know on what basis the authors estimate the differences between the two correlation coefficients and what kind of interaction the record in line 246-247 refers to. The discussion refers to the most important research findings. The concept of resilience as immunity shaped as a result of struggling with a difficult situation is similar to the concept of stress vaccination by Meichenbaum (2007). It is a pity that the discussion lacked a reference to this theory and therapy program. The bibliography is up-to-date and correct.

Author Response

Honored colleague,

We would like to thank you for your important and constructive feedback on our paper. All changes within the manuscript due to your suggestions are held in blue to make them easier to find.

We want to apologize for the slight of not having included the ethics committee statement, which we have now provided. We further added information on the investigated parameters in the MANCOVA in line 246-247. We also added a reference to the highly relevant stress vaccination concept that you pointed out and tried to integrate our findings within this context.

We hope to have addressed your input satisfactorily in our updated manuscript version.

Reviewer 4 Report

The study set out to investigate a question by examining whether workers in Austria are at the forefront of the fight against the virus. The study methods were adequate despite the small number of participants.  The authors used a questionnaire survey to elicit an appropriate level of questions.  The questionnaire could be used again in the future on a larger sample. Despite the small sample size, the research provides valuable and relevant results. The lack of representativeness also allows to draw appropriate conclusions. The methodological methods are appropriate and adapted to the small sample size. 

Author Response

Honored colleague,

We thank you for your kind feedback and are happy that you are satisfied with the quality and content of our study.